# Delta Neutrophil Index Does Not Differentiate Bacterial Infection without Bacteremia from Viral Infection in Pediatric Febrile Patients

**DOI:** 10.3390/children10010161

**Published:** 2023-01-14

**Authors:** Maro Kim, Jin Hee Lee, Young Ho Kwak, Hyun Kyung Kim, Hyuksool Kwon, Dongbum Suh, Do Kyun Kim, Ha Ni Lee, Jin Hee Kim, Jie Hee Jue, Soyun Hwang

**Affiliations:** 1Department of Emergency Medicine, Seoul National University Bundang Hospital, Seoul National University College of Medicine, Seoul 13620, Republic of Korea; 2Disaster Medicine Research Center, Seoul National University Medical Research Center, Seoul 03080, Republic of Korea; 3Department of Emergency Medicine, Seoul National University Hospital, Seoul National University College of Medicine, Seoul 03080, Republic of Korea; 4Department of Laboratory Medicine, Cancer Research Institute, Seoul National University College of Medicine, Seoul 03080, Republic of Korea; 5Department of Pediatrics, Yonsei University College of Medicine, Seoul 03722, Republic of Korea

**Keywords:** children, delta neutrophil index, bacterial infection, viral infection

## Abstract

Introduction: We sought to determine whether the delta neutrophil index (DNI), a marker that is reported to be used to predict the diagnosis, prognosis, and disease severity of bacteremia and sepsis, is useful in differentiating bacterial infection without bacteremia (BIWB) from viral infections (VI) in pediatric febrile patients in the emergency department (ED). Method: We conducted a retrospective analysis of febrile patients’ medical records from the pediatric ED of the teaching hospital. The patients with BIWB and those with VI were identified with a review of medical records. The primary outcome was the diagnostic performance of DNI in differentiating BIWB from VI. The secondary outcome was a comparison of the diagnostic performances of DNI, CRP, WBC, and neutrophil count between the two groups. Results: A total of 151 (26.3%) patients were in the BIWB group, and 423 (73.7%) were in the VI group. There was no significant difference in DNI between the two groups (3.51 ± 6.90 vs. 3.07 ± 5.82, mean ± SD, BIWB vs. VI). However, CRP levels were significantly higher in the BIWB group than in the VI group (4.56 ± 5.45 vs. 1.39 ± 2.12, mean ± SD, BIWB vs. VI, *p* < 0.05). The AUROCs of DNI, WBC count, neutrophil levels, RDW, and CRP levels were 0.5016, 0.5531, 0.5631, 0.5131, and 0.7389, respectively, and only CRP levels were helpful in differentiating BIWB from VI. Conclusion: In the absence of bacteremia, DNI would not be helpful in differentiating BIWB from VI in pediatric febrile patients.

## 1. Introduction

Fever is one of the main causes for children to visit the emergency department (ED) [1]. The proportion of pediatric patients with fever who visit the ED is reported to be approximately 14% in the United Kingdom, 25% in the United States, and 37.4% in South Korea [1,2,3]. The major cause of pediatric febrile patients is a viral infection that resolves spontaneously [4].

It is very important to distinguish between bacterial infections and viral infections in pediatric patients with fever. The tests performed to diagnose serious bacterial infection (SBI) include culture tests performed on peripheral blood, urine, and cerebrospinal fluid (CSF). However, they often take more than 48 hours to confirm the pathogen, most are invasive tests, and there are disadvantages such as contamination of the sample, which depends on the skill of the operator when collecting the sample [5,6,7]. On the other hand, biomarkers such as erythrocyte sedimentation rate (ESR), C-reactive protein (CRP), and procalcitonin are helpful in differentiating SBI and bacteremia from viral infections in pediatric patients with fever because the results can be confirmed quickly [8,9].

The delta neutrophil index (DNI) is an index indicating the fraction of immature granulocytes in the blood. It is a marker that has been recently reported to predict the diagnosis, prognosis, and disease severity of bacteremia and sepsis in adults and newborns [10,11,12,13,14].

The purpose of this study was to determine whether DNI was useful in differentiating bacterial infections without bacteremia (BIWB) from viral infections (VI) in pediatric febrile patients (0–18 years) who visited the ED.

## 2. Materials and Methods

### 2.1. Study Design and Setting

This study was conducted by a retrospective analysis of the medical records of patients who visited the pediatric ED of the teaching hospital, which is a tertiary care medical center and regional emergency center, from July 2015 to January 2016. The hospital includes 20,000 pediatric patients annually in the census, and the children who visited the pediatric ED were treated by the residents of the ED, who were supervised by pediatric emergency attending physicians. A clinical diagnosis was made by a PED physician after a careful examination and an evaluation of the patient’s primary laboratory or imaging studies. Thereafter, the patient was either discharged from the ED for outpatient care, monitored in the ED, or hospitalized.

### 2.2. Participants

We obtained a list of patients who underwent DNI testing among the pediatric patients who visited the pediatric ED during the period. DNI was calculated using an automatic hematology analyzer (ADVIA 120; Siemens Healthcare Diagnostics, IL, USA) as previously described [10]. Based on the medical records and laboratory and radiology test results, the patients were divided into a bacterial infection without bacteremia (BIWB) group and a viral infection (VI) group.

The BIWB group was defined as patients who tested positive for bacteria in cerebrospinal fluid, stool, or urine, or who had bacterial pneumonia (defined as a case of focal segment (e.g., bronchopneumonia) or lobar pulmonary opacity (e.g., lobar pneumonia) on a chest x-ray, which was read as bacterial pneumonia by a radiologist), an abscess or appendicitis confirmed by imaging studies, or cellulitis, tonsillitis, peritonsillar abscess, or scarlet fever confirmed according to the typical clinical symptoms and signs. The VI group included patients whose BI-related tests were negative, as well as patients who recovered without antibiotic treatment within 14 days among the patients who did not meet the criteria for bacterial infection, had a confirmed viral PCR test, or who had their disease confirmed according to typical clinical symptoms and signs [15].

The medical records and test results were reviewed by one researcher using a prewritten report form. After that, another researcher reviewed and confirmed it. We excluded patients who received an organ transplant, or had a chronic infection, immunodeficiency, chronic inflammatory disease, oncological disease, trauma, or coinfection with both a virus and a bacteria, and who did not meet the diagnostic criteria for bacterial and viral infection.

Patient age, sex, weight, body temperature, symptom duration, diagnosis at ED discharge, culture test results from the blood, CSF, or urine, antibiotic administration and type in the emergency department, chest X-ray or other imaging studies, CBC and differential count, CRP, procalcitonin, DNI, and disposition were collected

### 2.3. Outcome Measurement

The primary outcome was the diagnostic performance of DNI in differentiating BIWB group from VI group. The secondary outcome was a comparison of the diagnostic performances of DNI, CRP, WBC count, and neutrophil count between patients with bacterial infections and viral infections. In addition, for sensitivity analysis, the diagnostic performances of the biomarkers were analyzed to distinguish the urinary tract infection (UTI) group from the VI group.

### 2.4. Statistics

We performed multivariate logistic regression to confirm the difference between the two groups. Data are expressed as the mean and standard deviation or 95% confidence interval for continuous data and are presented as numbers and percentages for categorical variables. The diagnostic performance of the analysis based on serum biomarkers was evaluated using the area under the receiver operating characteristic curve (AUROC).

Since the criteria for BI are sometimes based on relatively subjective clinical findings, a sensitivity analysis was conducted to compare the UTI group diagnosed with objective urine culture results and the VI group.

Missing values were not replaced by other values, and patients with missing values were excluded. A *p* value less than 0.05 was considered statistically significant. All statistical analyses were performed using STATA/SE 14.2 software (StataCorp LP, College Station, TX, USA).

### 2.5. Ethics

This study was approved by the research ethics committee of the SNUH Institutional Review Board (IRB No. 2012-032-1179).

### 2.6. Results

During the study period, 1330 patients were discharged after visiting a PED and after having a DNI test. After excluding 756 patients, a total of 574 patients were included in the study (Figure 1). Of the 574 patients, 151 (26.3%) had a BIWB and 423 (73.7%) had a VI (Figure 1). There was no significant difference in sex, age, body temperature, and symptom duration between the two groups; the BIWB group had a longer hospital stay (Table 1). The clinical diagnosis at the time of discharge from the ED in each group is described in Table 1. All children with CNS infection and neonatal sepsis at discharge diagnosis were confirmed to have VIs. For other inflammatory markers, all 574 cases were included in the analysis because there were no missing values; in the case of CRP, only 551 cases were included in the analysis because there were 23 missing values.

### 2.7. Primary Outcome

There was no significant difference in DNI between the two groups (3.51 ± 6.90 vs. 3.07 ± 5.82, mean ± SD, BIWB vs. VI, *p* = 0.146) (Table 1). The AUROC of DNI was 0.5016, which did not help to differentiate between the two groups.

### 2.8. Secondary Outcome

There was no difference in WBC, neutrophil, or RDW between the two groups, and CRP was significantly higher in the BIWB group than in the VI group (4.56 ± 5.45 vs. 1.39 ± 2.12, mean ± SD, BIWB vs. VI, *p* ≤ 0.05) (Table 1). The AUROCs of WBC, neutrophil, RDW, and CRP were 0.5531, 0.5631, 0.5131, and 0.7389, respectively, and only CRP was helpful in differentiating BIWBs from VIs (Figure 2(1)).

### 2.9. Sensitivity Analysis

When comparing UTI (42 cases) and viral infection (423 cases), WBC and CRP were significantly higher in the UTI group than in the VI group, and there were no other differences in RDW and DNI (14,198 ± 6158 vs. 10,832 ± 5115 for WBC, 4.40 ± 3.64 vs. 1.39 ± 2.12 for CRP, mean ± SD, UTI vs. VI, *p* ≤ 0.05) (Table 2). The AUROCs of WBC, neutrophil, RDW, and CRP were 0.6751, 0.5806, 0.6532, and 0.8117, respectively (Figure 2(2)).

## 3. Discussion

As a result of the analysis of this study, in pediatric febrile patients, DNI was ineffective in differentiating patients with a VI from those with a BIWB, and only CRP showed good diagnostic performance among other inflammatory markers. To the best of our knowledge, this study is the first to analyze the ability of DNI to differentiate VI and BIWB in pediatric febrile patients.

In pediatric febrile patients, it is very important to differentiate infections caused by bacteria, which require immediate antibiotic treatment, from infections caused by viruses, which can be treated only by symptomatic treatment and observation. However, in some cases, the symptoms of bacterial infections overlap with the symptoms caused by viral infection, so it is not easy to differentiate these two etiologies. Numerous studies have been conducted to differentiate bacterial infections from viral infections in infants and children with fever [16,17,18].

CRP, procalcitonin, and WBC are the most studied inflammatory markers in pediatric febrile patients [8]. Among them, the diagnostic performance of CRP and procalcitonin were similar, and both tests showed the superior diagnostic performance to WBC. In meta-analysis, it was reported that procalcitonin performs better than leukocyte count and C-reactive protein for detecting serious bacterial infection among children with fever without source aged between 7days and 36 months. Overall sensitivity was 0.83 (95% CI 0.70 to 0.91) for procalcitonin, 0.74 (95% CI 0.65 to 0.82) for C-reactive protein, and 0.58 (95% CI 0.49 to 0.67) for leukocyte count. Overall specificity was 0.69 (95% CI 0.59 to 0.85) for procalcitonin, 0.76 (95% CI 0.70 to 0.81) for C-reactive protein, and 0.73 (95% CI 0.67 to 0.77) for leukocyte count [19]. In another meta-analysis targeting pediatric patients aged 1 month to less than 18 years of age with fever, it was reported that CRP and procalcitonin provided the most diagnostic value in finding serious infection among inflammatory markers [20].

DNI is an indicator of the proportion of immature granulocytes (IG) in the circulating blood, and there have been several reports that have shown it is helpful for assessing the prognosis and severity of sepsis in patients with suspected sepsis and to predict positivity in blood culture results [10,11,12,13]. However, most studies on DNI have focused on adult patients, and studies on children are limited to neonatal sepsis and bacteremia [14,21,22]. In our study, we confirmed that DNI was not helpful in distinguishing BIWB from VI, and only CRP was helpful. Alternatively, even in patients with UTI without bacteremia, DNI was not helpful in distinguishing patients from VI.

Based on the results of studies thus far, DNI can be helpful in detecting and predicting prognosis in patients with bacteremia or sepsis, but in the absence of bacteremia, since the DNI test may not be helpful, it seems necessary to consider using other inflammatory markers.

This study has several limitations. First, this study was conducted on patients who visited the pediatric emergency department of one hospital. Therefore, the results may be difficult to generalize depending on the clinical practice of the hospital. Second, since this study was conducted as a retrospective review of medical records, selection bias may occur or the diagnoses may not be accurate. Not all patients underwent the same type of test, culture, and PCR analysis, and in some cases, it was necessary to rely on the results of the treating physician’s examination to distinguish between bacterial and viral infection. However, to overcome this limitation, we established diagnostic criteria for bacterial and viral infections and tried to select patients based on those criteria. In the sensitivity analysis, patients with UTIs, which are definite bacterial infections, were analyzed by comparing this group with the viral infection group. Due to these limitations, a prospective study would be needed in the future to confirm the diagnostic ability of DNI for distinguishing BIWBs from VIs according to more objective test standards.

## 4. Conclusions

In the absence of bacteremia, DNI would not be helpful in differentiating bacterial infections from viral infections in pediatric febrile patients.

## Figures and Tables

**Figure 1 children-10-00161-f001:**
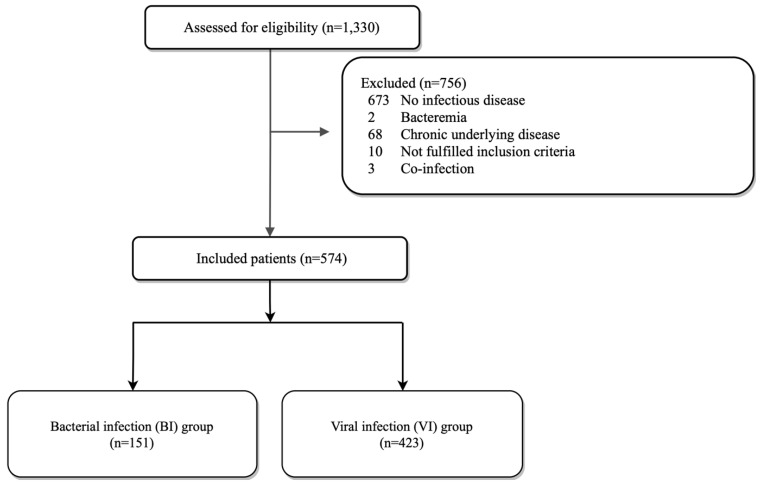
A total of 1330 patients were discharged after visiting a PED and having a DNI test performed. After excluding 756 patients, a total of 574 patients were included in the study. A total of 151 patients had bacterial infections, and 423 patients had viral infections. DNI: delta neutrophil index.

**Figure 2 children-10-00161-f002:**
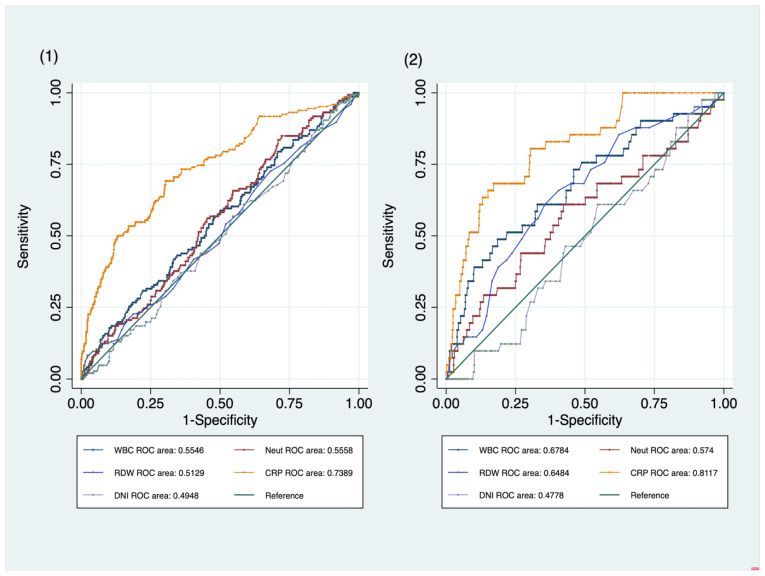
The AUROC of DNI was 0.4948, and the AUROC of CRP was 0.7389. The AUROC of WBC, neutrophils and RDW was 0.5546, 0.5558, and 0.5129, respectively. Only CRP was effective in distinguishing pediatric febrile patients with BIWB from those with VI (**1**). When comparing the patients with febrile UTI (42 cases) and the patients with VI (152 cases), WBC and CRP were significantly higher in the UTI group than in the VI group. The AUROCs of WBC, neutrophil, RDW, and CRP were 0.6751, 0.5806, 0.6532, and 0.8117, respectively. The AUROC of DNI was 0.4778, which did not help to differentiate between the two groups (**2**). AUROC: Area under ROC; DNI: Delta neutrophil index; CRP: C-reactive protein; WBC: White blood cell count; RDW: Red blood cell distribution width; BIWB: Bacterial infection without bacteremia; VI: Viral infection.

**Table 1 children-10-00161-t001:** Baseline demographic characteristics.

Characteristics	VI Group N = 423	BIWB Group N = 151	Odd Ratio	95% CI	*p* Value
Male, N (%)	251 (59.3)	84 (55.6)	1.55	0.92~2.63	0.098
Age (years), median (IQR)	2.83 (1.00~6.25)	3.92 (1.00~8.50)	1.04	0.97~1.11	0.227
Body temperature, mean ± SD	37.9 ± 0.1	38.1 ± 0.1	1.11	0.87~1.42	0.386
Symptom duration (days). mean ± SD	3.36 ± 0.13	4.02 ± 0.24	1.00	0.86~1.16	0.953
Hospital stays (days). median (IQR)	0 (0~1)	1 (0~3)	1.35	1.16~1.57	<0.05
White blood cell count, N/uL, mean ± SD	10,832 ± 5115	11,760 ± 5188	1.00	0.99~1.00	0.091
Neutrophil, N/uL median (IQR)	5770 (3580~9840)	6740 (4390~10,210)	0.99	0.99~1.00	0.129
RDW, %, mean ± SD	13.12 ± 0.98	13.20 ± 1.54	1.02	0.82~1.28	0.840
Delta neutrophil index, %	3.07 ± 5.82	3.51 ± 6.90	0.97	0.93~1.01	0.146
C-Reactive protein, mg/dL	1.39 ± 2.12	4.56 ± 5.45	1.28	1.17~1.41	<0.05
Diagnosis at ED, N (%)					<0.05
Upper respiratory infection	166 (39.2)	19 (12.6)			
Lower respiratory infection	43 (10.2)	43 (28.5)			
Gastrointestinal infection	131 (31.0)	37 (24.5)			
Genitourinary infection	0 (0.0)	42 (27.8)			
CNS infection	24 (5.7)	0 (0.0)			
Soft tissue infection	0 (0.0)	3 (2.0)			
Neonatal sepsis	12 (2.8)	0 (0.0)			
Other infection	47 (11.1)	7 (4.6)			

VI: Viral infection; BIWB: Bacterial infection without bacteremia; IQR: Interquartile range; RDW: Red blood cell distribution width; ED: emergency department.

**Table 2 children-10-00161-t002:** Sensitivity analysis. Comparison of each inflammatory marker between the UTI vs. viral infection groups.

Characteristics	VI Group (N= 423)	UTI Group (N = 42)	Odd Ratio	95% CI	*p* Value
Male, N (%)	251 (59.3)	26 (61.9)	1.55	0.92~2.63	0.098
Age (years), median (IQR)	2.83 (1.00~6.25)	0.38 (0.17~1.00)	1.04	0.97~1.11	0.227
Body temperature, mean ± SD	37.9 ± 0.1	38.5 ± 1.1	1.11	0.87~1.42	0.386
Symptom duration (days). mean ± SD	3.36 ± 0.13	2.55 ± 1.58	1.00	0.86~1.16	0.953
Hospital stays (days). median (IQR)	0 (0~1)	2 (1~3)	1.35	1.16~1.57	<0.05
White blood cell count, N/uL, mean ± SD	10,832 ± 5115	14,198 ± 6158	1.00	1.00~1.00	<0.05
Neutrophil, N/uL, median (IQR)	5770 (3580~9840)	7505 (3820~11,910)	0.99	0.99~0.99	<0.05
RDW, %, mean ± SD	13.12 ± 0.98	13.45 ± 2.07	1.01	0.80~1.48	0.599
Delta neutrophil index, %	3.07 ± 5.82	2.91 ± 3.19	0.97	0.91~1.03	0.278
C-Reactive protein, mg/dL	1.39 ± 2.12	4.40 ± 3.64	1.27	1.11~1.45	<0.05

VI: Viral infection; UTI: Urinary tract infection; IQR: Interquartile range; RDW: Red blood cell distribution width.

## Data Availability

The data sets analyzed during the current study are available from the corresponding author on reasonable request. Data can be provided after the article is published.

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
