# Peer review of "Delta Neutrophil Index Does Not Differentiate Bacterial Infection without Bacteremia from Viral Infection in Pediatric Febrile Patients"

_children, 2023, doi:10.3390/children10010161_

Round 1

Reviewer 1 Report

The authors assessed the ability of delta neutrophil index (DNI) to distinguish bacterial infection without bacteremia (BI) and viral infection. The diagnostic performance of DNI was compared to other know biomarkers using areas under the receiver operating characteristic (AUROCs).

Strengths: The article distinguishes its novelty by specifically targeting case of bacteremia with negative culture, a difficult category to manage. Sample size. Pediatrics cohort.

Limitations: Minor clarifications. Heterogeneity of the clinical assessments.

Comments:

          Were multivariate analysis/classification approaches tested?

          I recommend doing a regression model to take into account differences in clinical characteristics.

          Page 2, line 96: Clearly state the diagnostic criteria used for BI and VI.

          Line 224 “In the sensitivity analysis […], the purpose of the sentence is not clear.

          Line 226, The last sentence of the conclusion is vague and cast doubts on the tests used to diagnose BI and VI. I recommend reworking the Discussion/Conclusion: concisely summarize the study, avoid repeating results, and expand on the necessity for a better biomarker of sepsis in pediatrics that are faster than blood culture and independent of bacteremia.

          It can be confusing to talk about BI if only BI without infection was compared and tested. So, I suggested creating an acronym that will clearly differentiate those groups. E.g., BI without infection could be “BIWI”. Alternatively, the sentence could read like this: “ […] differentiating bacterial infection without bacteremia (hereafter referred to as BI) from viral infections (VI) […]”.

          Line 118, If missing value resulted in the exclusion of the patients, indicating how many remained for the auroc analysis.

Author Response

Thank you very much for your careful and valuable comments. Responses to comments have been added as attachments.

Reviewer 2 Report

This is a well written report investigating the biomarker delta neutrophil index (DNI). The study adds data on DNI:s performance in distinguishing viral and non bacteraemia bacterial infections in children with fever. I have just a few comments, however the question about the definition of bacterial and viral pneumonia is important as it is clinically difficult to distinguish these.

Regarding the title-please consider to changed it to the actual finding.

Line 38: The word ”Wound” should maybe be changed to “would”

In the material and method part I miss a clearer definition of pneumonia-how have you distinguished  viral and bacterial pneumonia?

Please re-formulate the following setting (line 199-202) as it could give the impression that you are talking about your own results. “Among them, the diagnostic performance of CRP and procalcitonin 199 was similar, and both tests showed superior diagnostic performance than WBC. The AU- 200 ROC achieved in the analysis of ED patients was 0.85 (95% CI 0.0.81–0.87) using CRP lev- 201 els and 0.84 (95% CI 0.81–0.87) using PCT levels.(19,20)”

Author Response

(The authors gave the same response as above.)
